# Prevalence and predictors of post-acute COVID syndrome among infected healthcare workers at University Malaya Medical Centre

Say Hiang Lim[1], Yin Cheng Lim[1]*, Rafdzah Ahmad Zaki[1,2], Bushra Megat Johari[3,4], Chung Yuan Chang[4], Sharifah Faridah Syed Omar[3,4], Amirah Azzeri[5,6], Maznah Dahlui[1,5,7,8], Adeeba Kamarulzaman[3,4,9]

1 Department of Social and Preventive Medicine, Faculty of Medicine, Universiti Malaya, Kuala Lumpur, Malaysia, 2 Centre for Epidemiology and Evidence-Based Practice, Faculty of Medicine, Universiti Malaya, Kuala Lumpur, Malaysia, 3 Department of Medicine, Faculty of Medicine, Universiti Malaya, Kuala Lumpur, Malaysia, 4 Centre of Excellent for Research in AIDS (CERIA), Faculty of Medicine, University of Malaya, Kuala Lumpur, Malaysia, 5 Department of Research Development and Innovation, University Malaya Medical Centre, Kuala Lumpur, Malaysia, 6 Community Health Unit, Department of Primary Care Faculty of Medicine and Health Sciences, Universiti Sains Islam Malaysia (USIM), Nilai, Malaysia, 7 Faculty of Medicine, University Taylor's Malaysia, Subang Jaya, Malaysia, 8 Faculty of Public Health, Universitas Airlangga, Surabaya, Indonesia, 9 Monash University Malaysia, Subang Jaya, Malaysia

* limyc@ummc.edu.my

**Data Availability Statement:** All relevant data are within the manuscript and its Supporting Information files.

## Abstract

### Background

Post Acute COVID Syndrome (PACS), a complex and poorly understood condition characterised by persistent symptoms following the acute phase of COVID-19 infection, has emerged as a significant global health concern. Healthcare workers who had been at the forefront of the pandemic response are at heightened risk of contracting the virus and subsequently developing PACS. Therefore, we aim to determine the prevalence and risk factors for PACS among healthcare workers infected with COVID-19.

### Methods

A cross-sectional study was conducted between October 2022 and August 2023 using an online REDCap electronic data capture tool questionnaire. PACS was defined as new or persistent symptoms lasting more than 28 days after a positive SARS-CoV-2 polymerase chain reaction or rapid test kit antigen test. Multivariable logistic regression was performed to determine predictors associated with PACS.

### Results

Among 609 infected healthcare workers, they were predominantly female (71.8%), Malays (84.6%), and aged 18–39 years (70.1%). 50.7% of infected healthcare workers experienced PACS. The most common PACS symptoms experienced were fatigue (27.9%), cough (25.1%), decreased physical strength (20.5%), and musculoskeletal pain (19.2%). Those who are more likely to develop PACS were females, underlying asthma, and COVID-19

**Funding:** This study is funded by Allianz Malaysia Berhad and its subsidiaries, Allianz Life Insurance Malaysia Berhad and Allianz General Insurance Company (Malaysia) Berhad (Research grant number: PV037-2021) for BMJ and AK. The URL of funder could be accessed at https://www.allianz.com.my/personal.htmThe funders had no role in study design, data collection and analysis, publication decisions, or manuscript preparation.

**Competing interests:** The authors have declared that no competing interests exist.

**Abbreviations:** BMI, Body Mass Index; COVID, Coronavirus Disease; OR, Odd Ratio; PACS, Post Acute COVID Syndrome; SPSS, Statistical Package for Social Science SPSS Statistics; UMMC, University Malaya Medical Centre.

severity category 3. On the other hand, those who received booster vaccinations were less likely to develop PACS.

## Conclusion

PACS is prevalent among healthcare workers with COVID-19 at the University Malaya Medical Centre. These findings emphasise the critical need for those with higher risk to receive regular health monitoring and checkups to detect any early signs of PACS. It underscores the need for continuous support and healthcare interventions to mitigate the impacts of PACS and ensure the physical and mental well-being of healthcare workers.

## Introduction

Coronavirus disease (COVID) is an infectious disease caused by the Severe Acute Respiratory Syndrome Coronavirus 2 (SARS-CoV-2) virus, which has disrupted healthcare services worldwide [1]. The first case of COVID -19 was first reported in Wuhan, China, in 2019 and subsequently spread worldwide [2]. In Malaysia, there were 5 million confirmed cases of COVID -19, with 37 thousand deaths as of July 2023 [3]. Persistence COVID -19 symptoms will lead to a condition with different terms used to address it, such as PACS, Long COVID, and post-acute sequelae of COVID -19, but what is of concern is the duration of the symptoms of COVID infection [4]. The prevalence of PACS could vary between different settings and populations because of differences such as the definition of PACS and the characteristics of the population that affect susceptibility [5]. Globally, according to the latest analysis, a pooled prevalence of PACS is between 9% and 63%, which is almost six times higher than other post-viral conditions such as postviral fatigue syndrome [6]. In Malaysia, a study done in Port Dickson described the prevalence of Long COVID symptoms at 27.4% [7]. The findings of the study done in Port Dickson were similar to another study in Malaysia, where the prevalence is 21.1% [8].

There were studies done that looked at persistent COVID symptoms in different timelines in Europe and Spain and described the prevalence of Long COVID after three weeks of infection at 52% [9]. The prevalence of PACS at 10–14 weeks post- COVID -19 infection was found to be 50.9% in another study in Spain [10]. Meanwhile, on the North American continent of the United States, research had been done to look at Long COVID prevalence at two months, which found the prevalence at 14.7% [11]. One study looked at four weeks and 12 weeks of Long COVID prevalence in the Asian continent of Bangladesh and found the prevalence of 4 weeks at 22.5% and PACS at 12 weeks at 16.1% [12]. Another study on the South American continent of Brazil found the prevalence of post-COVID-19 syndrome at 81% and 61% after 3 and 6 months after hospital discharge [13].

The long-term effects of COVID-19 may be due to damage at the cellular level in susceptible individuals. SARS-CoV-2 infection creates an inflammatory process with cytokine production that causes damage to the body's cells [14]. As a result of such a process, PACS could have both physical and mental effects on an infected person. Among the physical effects is musculoskeletal pain, which is increasingly reported as one of the most common persistent symptoms in PACS that could debilitate infected patients [15]. Shortness of breath is another common physical symptom because COVID-19 is principally a respiratory illness that could cause substantial damage to the respiratory tract via SARS-CoV-2 replications inside endothelial cells that cause damage to the cell lining, resulting in endothelial damage [16]. On the other hand,

the mental effects of PACS could be caused by inflammation in the brain, which had been linked with cognitive deficits and psychiatric manifestations [17, 18]. PACS is commonly associated with generalised anxiety, depression, post-traumatic stress disorder (PTSD), and sleep disturbance.

Healthcare workers are the most vulnerable groups to COVID-19 infection because of the higher risk of exposure. Studies in Malaysia have shown that more than half (53%) of COVID -19 infections among healthcare workers happen in the workplace, with 17% occurring between patients and staff [19]. This is in line with another study that evaluates the prevalence of COVID -19 infection among healthcare workers in Malaysia, which stood at 17.4% [20]. On the other hand, the prevalence of post COVID symptoms among healthcare workers in Hospital Kuala Lumpur is 51.4%, albeit the definition of post COVID symptoms in that study was not reported [21]. A separate study conducted in Malaysia showed that the nursing profession occupational group (40.5%) had the highest rate of COVID infection, followed by medical doctors (24.1%), healthcare assistants (9.7%), medical doctor assistants (9.1%), and administrative personnel (3.4%) [19]. Another study in Switzerland showed over half of the healthcare workers, or 309 (50.7%), developed Post COVID conditions [22]. The prevalence of both studies in Malaysia and Switzerland is higher compared to another study done in India among healthcare workers, which shows a 30.34% prevalence of Long COVID [23]. These facts imply a high rate of PACS among healthcare workers and could affect their work performance. Evidence shows that COVID-19 may be associated with long-term health consequences affecting employees' work performances [24]. These may later affect their work by increasing their absence from work (absenteeism) and decreasing their productivity while working (presenteeism) [25].

As the paradigm of COVID-19 shifts to understanding its long-term effects and previous research in Malaysia has made significant findings in understanding the long-term effects of COVID-19, a notable gap exists in the limited focus on healthcare workers, who constitute a unique and high-risk group. Previous studies have predominantly concentrated on the general population, leaving the specific patterns of PACS among healthcare workers relatively unexplored [7, 8]. This gap in the literature is significant because healthcare workers were at the forefront during and after the pandemic, facing increased exposure to the virus and unique occupational stressors. Their experiences and risks may differ from those of the general population, potentially leading to distinct patterns of PACS. Therefore, our study aims to address this knowledge gap by investigating the factors associated with PACS among healthcare workers. By doing so, we aim to provide valuable insights that can inform preventive measures and ensure the preservation of adequate staffing and work shifts in healthcare settings.

## Methods

### Study design and setting

This was an analytical cross-sectional study of the prevalence of Post Acute COVID Syndrome among healthcare workers infected by COVID-19 between 1st January 2020 and 1st August 2023. This study was conducted between October 2022 and August 2023 at the University Malaya Medical Centre (UMMC), situated strategically in Kuala Lumpur and Selangor (Klang Valley). UMMC is a tertiary hospital with 11 multidisciplinary specialties. As of 2018, UMMC had 5844 employees, of which 5397 were permanent and 447 were contract-based. UMMC is also a COVID referral centre, implying a higher risk of infection among healthcare workers [26].

### Data collection

Data was collected using an online REDCap electronic data capture tool questionnaire. All eligible participants with a history of COVID-19 infection between 1st January 2020 and 1st

August 2023 were recruited. We defined a COVID-19 patient as a person with a laboratory confirmation for SARS-CoV-2 by reverse transcriptase-polymerase chain reaction (RT-PCR) or Rapid Test Kit Antigen(RTK-Ag) test [27]. Data was collected using a redcap link with the online questionnaires sent via email through the human resources department and WhatsApp applications with the list of infected healthcare workers obtained from the UMMC public health surveillance record.

## Dependent variable

The primary outcome of this study was the occurrence of any PACS symptoms among infected healthcare workers. PACS was defined as any continuation of COVID-19 symptoms after 28 days [28]. The list of post-COVID symptoms was adapted from the National Health Service (NHS), which assesses patients with post-COVID infection and determines their persistent symptoms [29]. There were a total of 28 symptoms that were classified according to systems. Respiratory system symptoms include shortness of breath and cough. Cardiovascular system symptoms include palpitation and chest discomfort. Nervous system symptoms include brain fog, difficulty concentrating, sleep disturbance, and visual disturbance. Musculoskeletal system symptoms include decreased physical strength, musculoskeletal pain, and base of skull pain. Psychological system symptoms include lack of motivation, low mood, anxiety, feeling down, nightmares, and feeling depressed. Gastrointestinal system symptoms include nausea. Reproductive system symptoms include loss of sexual interest. Ear, nose, and throat system symptoms include tinnitus, loss of smell, and loss of taste. Generalised symptoms include fatigue, headache, rashes, recurrent fever, and loss of weight.

## Independent variables

The baseline characteristic of this study, such as ages, genders, ethnicities, smoking status, body weights, heights, past medical histories, vaccination statuses, COVID-19 infection history, and acute COVID-19 symptoms, were self-reported by healthcare workers. The ethnic groups were categorised as Malays, Chinese, Indians, and other ethnicities in this multiracial country. Smoking status was yes or no to current smoking. Body mass index (BMI) was computed using REDCap after participants provided their height and weight. BMI is categorised following the Ministry of Health Malaysia guidelines as below 18.5 is underweight, 18.5–24.9 is normal weight, 25–29.9 is overweight, and above 30 is obese [30]. The comorbidities included diabetes, hypertension, cardiovascular disease (CVD), asthma, chronic obstructive airway disease (COAD), neoplasm and other underlying illnesses. COVID-19 reinfection history is categorised as first, second, and more than three times infection, as there are no limits on how many reinfections can occur [31]. COVID vaccination status is categorised as complete primary dose, single dose, or not vaccinated. Vaccination booster is categorised as one dose, two doses, or does not receive any booster dose. COVID-19 infection severity is categorised as Category 1 (No symptoms), Category 2 (Mild Symptoms), Category 3 (Symptoms with Pneumonia), Category 4 (Category 3 with needed oxygen support), Category 5 (Critically ill with organ involvement) [32]. Duration since COVID-19 infection is categorised as one to three months, three to six months, and above six months.

## Sample size

The sample size was calculated using the following formula: n = Z 2 P(1−P) / d2, where n is the sample size, Z is the statistic corresponding to a level of confidence, and P is the expected prevalence, which in this study was estimated to be moderate at 51% and d is precision (corresponding to effect size) and was selected to be 5%. For an unknown population size (infinite

population assumed), the expected sample size was 384. With an expected attrition rate of 10%, we intend to recruit 420 eligible participants.

## Statistical analysis

Descriptive analysis was performed for the baseline characteristics and PACS symptoms. The variables were presented as mean ± standard deviation (SD) for normally distributed data, while median and inter-quartile range (IQR) were presented for skewed data. Both frequency and percentages were reported for categorical variables. Univariate binary logistic regression was first conducted to determine individual factors associated with PACS. Multivariate logistic regression analysis was carried out for variables with p-value <0.25 and clinically essential variables. We reported the adjusted odd ratios with 95% confidence intervals. All the tests were two-tailed, and statistical significance was fixed at p-value<0.05. All analyses were done using IBM SPSS Statistical Software, version 27.

## Ethical considerations

Ethical approval was obtained from the Medical Research Ethics Committee (MREC) University of Malaya Medical Centre, identification number 2021325–9983 and 2022104–11593. In adherence to ethical guidelines, digital informed consent was sought from all participants involved in this study. Participants who agreed to participate in this study were sent informed consent approval and details regarding this study. All respondents were assured of confidentiality.

## Results

A total of 609 eligible participants participated in the current study. Among the respondents, the median age was 36, with (73.7%) below 40 years old. Most respondents were female (73.4%), mainly Malay (84.6%). 240 (39.4%) of respondents had a normal weight, 194 (31.9%) were overweight, and 147 (24.1%) were obese.

The common comorbidities among the respondents were asthma (7.9%), hypertension (7.6%), and diabetes mellitus (3.9%). On the other hand, 424 (69.6%) respondents had no known comorbidities. Most respondents didn't smoke (98.2%). Most infected healthcare workers were categorised as having mild symptoms (85.1%), and 591 (97%) had completed their primary vaccination. On top of it, 402 (68.1%) had received at least one booster vaccine, and 143 (24.1%) of infected healthcare workers had received a second COVID-19 booster vaccination. The prevalence of persistent symptoms above four weeks was (50.7%), above three months (31.4%), and above six months (20.9%). The highest PACS symptoms above four weeks were fatigue (27.9%), cough (25.1%), and decreased physical strength (20.5%). In addition, PACS symptoms most common after three months were fatigue (16.24%), decreased physical strength (11.5%), and musculoskeletal pain (11.2%). PACS symptoms that persist after six months are fatigue (9.2%), musculoskeletal pain (6.7%), and brain fog (5.9%). The duration of each symptom is further illustrated as a bar chart in **Fig 1.**

The demographic and medical characteristics of infected healthcare workers with and without PACS are illustrated in **Table 1**. Among infected healthcare workers with PACS, (79.3%) were female in comparison with only (64%) among those without PACS. Infected healthcare workers with underlying asthma who developed PACS (10.4%) were also two times more likely than those without PACS (5.3%). Those with PACS also had a higher percentage among those with category 2 COVID severity (87.1%) and category 3 (3.2%) compared with those without PACS (83%) and (1%), respectively. In addition, those without PACS had a higher percentage

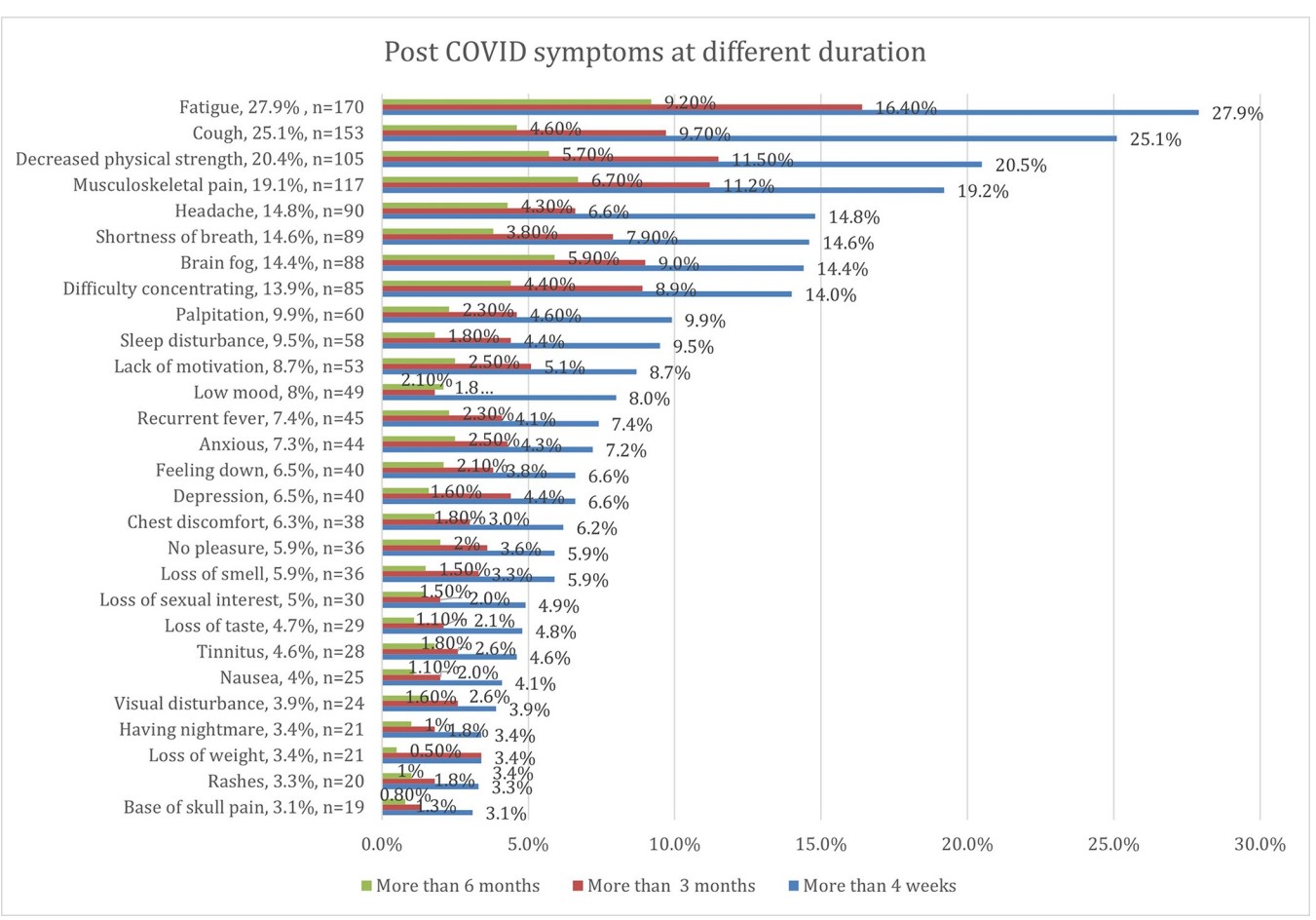

**Fig 1. Post-COVID commonest symptoms in descending order against duration.**

among those who received the first vaccination booster (68.6%) and second vaccination booster (25.3%) compared with those with PACS (67.4%) and (23.1%), respectively.

Simple logistic regression was performed (Table 2), and variables with p-value <0.25, which were gender, underlying asthma, COVID severity infection, and vaccination booster, were selected to be included in the final model.

In the final multiple logistic regression model, four independent factors were associated with PACS: females, underlying asthma, COVID severity category 3, and vaccination boosters (Table 3). The associated factor with the highest OR is COVID severity category 3 (aOR:4.85, 95% CI 1.21–19.45). Underlying asthma disease and the female gender were twice as likely to develop PACS as their counterparts. Regardless of one or two doses, vaccination boosters protect from developing PACS.

## Discussion

In this study, over half of the HCWs, or 309 (50.7%), had PACS, with the most common symptoms being fatigue (27.9%), cough (25.1%), and decreased physical strength (20.5%). The risk factors that were found to be associated with PACS are female gender, underlying asthma, COVID-19 infection severity, and COVID-19 vaccination booster status.

The prevalence of healthcare workers developing PACS is similar to another study conducted in the Hospital Kuala Lumpur Nephrology Department, Malaysia, where 51.4% [21]

**Table 1. Characteristics between healthcare workers with Post Acute COVID Syndrome (PACS) and without PACS.**

| Characteristics | Total (n = 609) | Without PACS (n = 300) | With PACS (n = 309) | |
|---|---|---|---|---|
| | n (%) | n (%) | n (%) | $\chi^2$ (p-value) |
| **Age (years)** | | | | 0.37 |
| 18–39 | 449(74%) | 225(75.0%) | 224(72.5%) | |
| 40–60 | 160(26%) | 75(25.0%) | 85(27.5%) | |
| **Gender** | | | | <0.05 |
| Male | 162(26.6%) | 98(32.7%) | 64(20.7%) | |
| Female | 447(73.4%) | 202(67.3%) | 245(79.3%) | |
| **Ethnicity** | | | | 0.83 |
| Malay | 515(84.6%) | 253(84.3%) | 262(84.8%) | |
| Chinese | 56(9.2%) | 30(10.0%) | 26(8.4%) | |
| Indian | 35(5.7%) | 16(5.3%) | 19(6.1%) | |
| Others | 3(0.5%) | 1(0.3%) | 2(0.6%) | |
| **Body Mass Index** | | | | <0.05 |
| <18.5 | 28(4.6%) | 16(5.3%) | 12(3.9%) | |
| 18.5–24.9 | 240(39.4%) | 132(44.0%) | 108(35.0%) | |
| 25–29.9 | 194(31.8%) | 93(31.0%) | 101(32.6%) | |
| >30 | 147(24.1%) | 59(19.7%) | 88(28.5%) | |
| **Diabetes Mellitus** | | | | 0.45 |
| No | 585(96%) | 290(96.7%) | 295(95.5%) | |
| Yes | 24(4%) | 10(3.3%) | 14(4.5%) | |
| **Hypertension** | | | | 0.26 |
| No | 563(92.4%) | 281(93.7%) | 282(91.3%) | |
| Yes | 46(7.6%) | 19(6.3%) | 27(8.7%) | |
| **Ischemic Heart Disease** | | | | 0.32 |
| No | 608(99.8%) | 300(100%) | 308(99.7%) | |
| Yes | 1(0.2%) | 0(0%) | 1(0.3%) | |
| **Asthma** | | | | <0.05 |
| No | 561(92.1%) | 284(94.7%) | 277(89.6%) | |
| Yes | 48(7.9%) | 16(5.3%) | 32(10.4%) | |
| **HIV** | | | | 0.32 |
| No | 608(99.8%) | 300(100%) | 308(99.7%) | |
| Yes | 1(0.2%) | 0(0%) | 1(0.3%) | |
| **Malignancy** | | | | 0.32 |
| No | 608(99.8%) | 300(100%) | 308(99.7%) | |
| Yes | 1(0.2%) | 0(0%) | 1(0.3%) | |
| **Smoking status** | | | | 0.80 |
| No | 598(98.1%) | 295(98.3%) | 303(98.1%) | |
| Yes | 11(1.9%) | 5(1.7%) | 6(1.9%) | |
| **Time since COVID infection(months)** | | | | 0.61 |
| 1–3 | 58(9.5%) | 27(9%) | 31 10%) | |
| >3–6 | 29(4.7%) | 12(4%) | 17(5.5.%) | |
| >6 | 522(85.8%) | 261(87%) | 261(84.5%) | |
| **COVID infection frequency** | | | | <0.05 |
| 1 | 421(69.1%) | 219(73%) | 202(65.4%) | |
| 2 | 159(26.1%) | 68(22.7%) | 91(29.4%) | |
| ≥3 | 29(4.8%) | 13(4.3%) | 16(5.2%) | |

(*Continued*)

**Table 1.** (Continued)

| Characteristics | Total (n = 609) | Without PACS (n = 300) | With PACS (n = 309) | |
|---|---|---|---|---|
| | n (%) | n (%) | n (%) | χ2 (p-value) |
| **COVID vaccination status** | | | | 0.59 |
| Completed Primary | 591(97.0%) | 290(96.7%) | 301(97.4%) | |
| Never had vaccine | 18(3%) | 10(3.3%) | 8(2.6%) | |
| **Booster dose*** | | | | 0.21 |
| Never had booster | 46(7.8%) | 18(6.1%) | 28(9.4%) | |
| 1 | 402(68.1%) | 201(68.6%) | 201(67.4%) | |
| 2 | 143(24.1%) | 74(25.3%) | 69(23.1%) | |
| **COVID severity(category)** | | | | <0.05 |
| 1 | 78(12.8%) | 48(16.0%) | 30(9.7%) | |
| 2 | 518(85.1%) | 249(83.0%) | 269(87.1%) | |
| 3 | 13(2.1%) | 3(1.0%) | 10(3.2%) | |

*Total n for booster dose respondents without PACS is 290, and with PACS is 301 as some never had vaccination; thus, it does not sum to a total n of 300 and 309, respectively. The total n for both categories is 591.

have long-term effects of COVID-19. The prevalence found in this study is comparably higher than the study conducted in the general population of Malaysia. The study in Port Dickson reported that 27.4% of respondents still have COVID-19 symptoms after four weeks [7]. Another cross-sectional study of Malaysia's general population reported that 48% of respondents still have persistent symptoms after six weeks [8]. Globally, there are other studies with similar findings of PACS prevalence, as in one systematic study of PACS, where 51% have it [33]. A similar systematic review found that the general population's PACS prevalence is 56.9% [34].

This study findings also show a higher risk of developing PACS compared with another study done in Brazil and India among healthcare workers, which showed a prevalence of 27.4% and 30.34%, respectively [23, 35]. The prevalence difference between settings could be due to the study in Brazil involving a multicenter and a larger sample of 7051 healthcare workers. At the same time, the study in India also consists of eight tertiary care centres. Aside from that, this could also be due to differences in individual susceptibility to PACS [36]. Aside from that, studies that involve healthcare workers could have a higher prevalence of PACS because of an increase of almost ten times the infectivity rate, as described in a large-scale study involving around 2.8 million respondents conducted in the UK and the US [37].

The most common symptoms that persist four weeks after the infection in this study are fatigue (27.9%) and cough (25.1%) are similar to the study conducted in Hospital Kuala Lumpur (HKL), Malaysia, which reported the most common symptoms as fatigue (22.9%), chest pain and palpitation (13.4%), and cough and shortness of breath (12.7%) [21]. However, the difference in the prevalence of this study compared to the study done in HKL could be because the study consists of only healthcare workers from the nephrology department. In contrast, this study includes all healthcare workers in the hospital. Similarly, the study in Port Dickson, Malaysia, also found fatigue and cough are among the most common symptoms, with 54.0% and 20.2% each having it [7]. On top of that, this study common PACS symptoms are also similar to those reported in another study [38], which are fatigue (45.2%), musculoskeletal pain (38.2%), and loss of smell (35.0%). The difference in prevalence between this study and that study, however, could be due to the different population settings where, in that study, a national base study is conducted with a larger sample size. The fact that this study found

**Table 2. Univariable analysis of factors associated with Post Acute COVID Syndrome.**

| Variables | Crude OR | 95% CI | | p-value |
|---|---|---|---|---|
| | | Lower | Upper | |
| **Age (years)** | | | | |
| 18–39 | Reference | Reference | | |
| 40–60 | 1.18 | 0.82 | 1.70 | 0.37 |
| **Gender** | | | | |
| Male | Reference | Reference | | |
| Female | 1.85 | 1.29 | 2.68 | <0.01* |
| **Ethnicity** | | | | |
| Malay | Reference | Reference | | |
| Chinese | 0.52 | 0.05 | 5.75 | 0.59 |
| Indian | 0.43 | 0.04 | 5.06 | 0.50 |
| Others | 0.59 | 0.05 | 7.17 | 0.68 |
| **Body Mass Index** | | | | |
| <18.5 | Reference | Reference | | |
| 18.5–24.9 | 1.09 | 0.50 | 2.41 | 0.83 |
| 25–29.9 | 1.45 | 0.65 | 3.22 | 0.36 |
| >30 | 2.00 | 0.85 | 4.51 | 0.10 |
| **Diabetes Mellitus** | | | | |
| No | Reference | Reference | | |
| Yes | 1.38 | 0.60 | 3.15 | 0.45 |
| **Hypertension** | | | | |
| No | Reference | Reference | | |
| Yes | 1.42 | 0.77 | 2.61 | 0.26 |
| **Asthma** | | | | |
| No | Reference | Reference | | |
| Yes | 2.05 | 1.10 | 3.82 | 0.024* |
| **Alcohol Consumption** | | | | |
| No | Reference | Reference | | |
| Yes | 0.65 | 0.11 | 3.89 | 0.63 |
| **Smoking** | | | | |
| No | Reference | Reference | | |
| Yes | 1.17 | 0.35 | 3.87 | 0.80 |
| **Time since COVID infection(months)** | | | | |
| 1 | Reference | Reference | | |
| >3–6 | 1.23 | 0.50 | 3.04 | 0.65 |
| >6 | 0.87 | 0.51 | 1.50 | 0.62 |
| **COVID infection frequency(time)** | | | | |
| 1 | Reference | Reference | | |
| 2 | 1.45 | 1.00 | 2.10 | 0.05 |
| ≥3 | 1.33 | 0.63 | 2.84 | 0.46 |
| **Vaccination status** | | | | |
| No | Reference | Reference | | |
| Yes | 0.77 | 0.30 | 1.98 | 0.59 |
| **Vaccination booster** | | | | |
| No | Reference | Reference | | |
| 1 | 0.64 | 0.37 | 1.10 | 0.11* |
| 2 | 0.60 | 0.33 | 1.09 | 0.09* |

(*Continued*)

**Table 2.** (Continued)

| Variables | Crude OR | 95% CI | | p-value |
|---|---|---|---|---|
| | | Lower | Upper | |
| **COVID severity(category)** | | | | |
| 1 | Reference | Reference | | |
| 2 | 1.73 | 1.06 | 2.82 | 0.03* |
| 3 | 5.33 | 1.36 | 20.96 | 0.02* |

*Notes*: *p-value<0.25, statistical significance to perform multiple logistic regression

infected healthcare workers with PACS having a high proportion of fatigue and cough is worrying because it could affect their work performance and daily activities, including their quality of life.

This study shows that there was an association between the female gender with an aOR of 1.77 (95%CI: 1.22–2.97) in developing PACS. Female gender as a risk factor for PACS was also found in studies conducted among general populations. In India, a study shows females had an OR of 1.78(95%CI: 1.09–2.26) [39] to develop PACS. In addition, in another study conducted in Iran, females also had Long COVID more frequently than men (OR: 1.268; 95% CI: 1.122–1.432) [40]. Similarly, other findings in the United Kingdom describe that the female gender had an OR of 1.56 (95% CI, 1.41–1.73) [41] to develop post- COVID conditions.

Meanwhile, in Brazil, a study conducted among healthcare workers also described females as a risk factor for Long COVID with an OR of 1.21(95% CI 1.05–1.39) [42]. The female gender had a higher odds of developing PACS as biological sex could influence susceptibility to infection, diseases, and outcomes. Studies also explain that the female sex has more robust innate and adaptive immune responses, which could hinder them from initial infection but render females more vulnerable to prolonged autoimmune-related diseases [43]. Another study also states that the X chromosome found in females could reduce the susceptibility of

**Table 3. Multivariable analysis of factors associated with Post Acute COVID Syndrome.**

| Variables | Crude OR (95% CI) | p-value | Adjusted OR (95% CI) | p-value |
|---|---|---|---|---|
| **Gender** | | | | <0.05 |
| Male | Reference | | Reference | |
| Female | 1.85(1.29–2.68) | <0.01 | 1.77(1.22–2.97) | |
| **Asthma** | | | | <0.05 |
| No | Reference | | Reference | |
| Yes | 2.05(1.10–3.82) | 0.024 | 2.01(1.07–3.78) | |
| **COVID severity(category)** | | | | 0.053 |
| 1 | Reference | | Reference | |
| 2 | 1.73(1.06–2.82) | 0.028 | 1.64(1.00–2.72) | <0.05 |
| 3 | 5.33(1.36–20.96) | 0.017 | 4.85(1.21–19.45) | |
| **Vaccination booster** | | | | |
| No | Reference | | Reference | |
| 1 | 0.64(0.37–1.10) | 0.106 | 0.57(0.33–1.00) | <0.05 |
| 2 | 0.60(0.33–1.09) | 0.093 | 0.50(0.27–0.94) | <0.05 |

*Notes*: The adjusted odds ratio (95% CI and p-value) was computed in the multivariable logistic regression model, adjusted for gender, asthma comorbidities, COVID Severity, and vaccination booster.

females to viral infection but would increase the inflammatory process when infected, which is attributed to PACS [44]. The differences in OR of female gender between this study and others could be because of different settings where working in hospitals could result in a higher risk of contracting COVID-19 and increase the risk of PACS [37].

Among the comorbidities analysed, asthma is the only significant risk factor for developing PACS in general, irrespective of symptoms, with an aOR of 2.01. Asthma as a risk factor for Long COVID was also found in a study in Poland (OR 1.56, 95% CI 1.01–2.41) [45]. In addition, a study in Germany also found that those with comorbidity asthma are at higher risk of developing post-COVID conditions (OR 1.38 95%CI 1.19–1.59) [46]. One study in the United Kingdom also found asthma was the only preexisting condition significantly associated with Long COVID (OR = 2.14, 95% CI: 1.55–2.96) [47]. Asthma has a higher risk of developing PACS as asthmatics usually have an increased susceptibility to common viral respiratory infections [48]. Asthma also increases the risk of PACS as it is a chronic inflammatory lung disease, and COVID-19 also primarily affects the upper and lower airways, leading to marked inflammation [49]. The difference in OR among studies could occur as the other study does not include the severity of COVID-19 and vaccination status as confounders, which could influence the end outcome of PACS. Including the severity and vaccination status of infected COVID-19 patients as confounders is essential because these factors could improve or worsen PACS.

COVID severity is another significant factor in this study, with an aOR 4.85 (95% CI: 1.21–19.45) in developing PACS for healthcare workers with Category 3 COVID-19 infection. This study findings that COVID severity as a risk factor for PACS is also found in other studies among the general population in Poland, with an OR of 2.27 (95% CI: 1.82–2.83) [50], and the analysis done in the Mediterranean, with an OR of 2.87 (95% CI:1.13–7.32) [10]. The CDC of the United States has stated that groups with severe COVID-19 infection are more likely to have lasting COVID-19 sequelae due to multiorgan damage [28]. COVID severity could lead to a higher risk of developing PACS, which could be caused by high serum cortisol levels and proinflammatory cytokines that persist after acute infection [51]. The differences could occur because of different study designs, as the study in Poland analyses data from the registry.

In contrast, studies in the Mediterranean utilised cohort study design and followed patients for two months. In this study, among the predictors of protective factors for developing PACS were receiving vaccination boosters irrespective of one or two doses. The odds of developing PACS after receiving one and two doses are 0.57 and 0.50, respectively. The protective effect of vaccine boosters was also found in a study conducted in Italy, with an OR of 0.16 [52]. COVID-19 vaccination could prevent COVID-19 infection and reduce the infection severity, reducing the risk of developing Long COVID [53]. The difference in findings could be because of the different timelines for receiving vaccines, as the vaccine effect could diminish as time passes.

## Strength and limitation

To the best of our knowledge, this was the first study on the predictors for PACS among healthcare workers with COVID-19 infection in Malaysia. Therefore, the findings provide an overview of the impact of COVID-19 on our healthcare workers in hospital settings. This study is also conducted at UMMC, a referral centre for COVID-19 that has exposure to a larger patient population that increases the risk of infection among healthcare workers, which can be valuable for collecting data and conducting research on COVID-19. Aside from that, this study uses an online questionnaire to gather the data needed. Participants could answer at their convenience as specific groups of healthcare workers work in shifts, and gathering data

through online surveys is more robust. The limitation of this study is that it is conducted in a single setting where different settings could produce an additional yield. Furthermore, the cross-sectional study design in this research did not allow for the establishment of a temporal relationship between exposure and outcome. Likewise, the cross-sectional study design also permits recall bias among respondents.

Moreover, we did not collect data on the type of vaccine HCWs took, which could produce different levels of protection against PACS. The other limitation of this study is that we did not collect data on the occupation of HCWs, which could have yielded further findings. Therefore, future research could benefit from considering such data for a more comprehensive analysis of healthcare workforce dynamics.

## Conclusion

In conclusion, over half of healthcare workers with COVID-19 in UMMC, Malaysia, experienced PACS symptoms. The findings of this study highlight the importance of monitoring and addressing the long-term health effects of COVID-19 among this crucial workforce. It underscores the need for ongoing support, resources, and healthcare interventions to mitigate the impacts of PACS and ensure the physical and mental well-being of healthcare workers. Significant consideration should be directed towards the implementation of targeted interventions for high-risk healthcare workers who are susceptible to PACS. The study findings indicate that healthcare workers who received vaccination boosters exhibited a protective effect of PACS, highlighting the importance of policymakers prioritising and actively encouraging them to receive vaccination boosters.

## Supporting information

**S1 Data.**
(XLSX)

## Acknowledgments

The Post Acute COVID symptoms study (PACS study) was an initiative by multidisciplinary departments under the University Malaya Medical Centre, Malaysia. The authors appreciate the efforts contributed by the team members of Social and Preventive Medicine, Infectious Diseases, Geriatrics, Primary Care, Rehabilitation, Respiratory, and Research Development and Innovation. In addition, the authors are most grateful to all the participants and the research coordinators for their contributions to this study.

## Author Contributions

**Conceptualization:** Yin Cheng Lim, Rafdzah Ahmad Zaki, Bushra Megat Johari, Adeeba Kamarulzaman.

**Data curation:** Say Hiang Lim.

**Formal analysis:** Say Hiang Lim, Maznah Dahlui.

**Funding acquisition:** Bushra Megat Johari, Adeeba Kamarulzaman.

**Methodology:** Yin Cheng Lim, Rafdzah Ahmad Zaki, Amirah Azzeri, Maznah Dahlui.

**Supervision:** Chung Yuan Chang.

**Writing – original draft:** Say Hiang Lim.

**Writing – review & editing:** Say Hiang Lim, Yin Cheng Lim, Rafdzah Ahmad Zaki, Bushra Megat Johari, Chung Yuan Chang, Sharifah Faridah Syed Omar, Amirah Azzeri, Maznah Dahlui, Adeeba Kamarulzaman.

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
