## [Decision Letter · Decision Letter 0]

4 Jan 2024

PONE-D-23-35985Prevalence and Predictors of Post-Acute Covid Syndrome Among Infected Healthcare Workers at University Malaya Medical CentrePLOS ONE

Dear Dr. Lim,

Thank you for submitting your manuscript to PLOS ONE. After careful consideration, we feel that it has merit but does not fully meet PLOS ONE’s publication criteria as it currently stands. Therefore, we invite you to submit a revised version of the manuscript that addresses the points raised during the review process.

**ACADEMIC EDITOR: Thank you for submitting your manuscript to PLoS One. Please find below the reviewer comments and address them so that you manuscript may be suitable for publication here.**==============================

We look forward to receiving your revised manuscript.

Kind regards,

Belal Nedal Sabbah

Academic Editor

PLOS ONE

5. PLOS requires an ORCID iD for the corresponding author in Editorial Manager on papers submitted after December 6th, 2016. Please ensure that you have an ORCID iD and that it is validated in Editorial Manager. To do this, go to ‘Update my Information’ (in the upper left-hand corner of the main menu), and click on the Fetch/Validate link next to the ORCID field. This will take you to the ORCID site and allow you to create a new iD or authenticate a pre-existing iD in Editorial Manager. Please see the following video for instructions on linking an ORCID iD to your Editorial Manager account: https://www.youtube.com/watch?v=_xcclfuvtxQ.

6. We are unable to open your Supporting Information file [Manuscript Data.sav]. Please kindly revise as necessary and re-upload.

Reviewers' comments:

Reviewer's Responses to Questions

**Comments to the Author**

1. Is the manuscript technically sound, and do the data support the conclusions?

Reviewer #1: Yes

Reviewer #2: Yes

2. Has the statistical analysis been performed appropriately and rigorously? 

Reviewer #1: Yes

Reviewer #2: Yes

3. Have the authors made all data underlying the findings in their manuscript fully available?

Reviewer #1: Yes

Reviewer #2: Yes

4. Is the manuscript presented in an intelligible fashion and written in standard English?

Reviewer #1: No

Reviewer #2: Yes

5. Review Comments to the Author

Reviewer #1: This is a well-structured manuscript which presents findings from a cross-sectional study that investigates prevalence of post-acute covid syndrome and its associated risk factors among healthcare workers in a large COVID referral hospital in Malaysia. The manuscript is simple and captures the necessary information well. it is structured accordingly and easy to read. Conclusions are sound and based on findings from the study. Findings add to the existing body of knowledge.

A couple of areas for improvement as follows:

1. There are some grammatical errors in the paper. Authors use past tense, present tense and occasional future tense throughout the paper, especially in reporting results. Please review.

2. Line 91-93 sounds incomplete.

3. Line 232 in the results section, figure 1 is referred to but it is only available as an additional resource. It would be helpful to the reader if the figure is availed in the paper just like the tables.

4. Correct line 233 to reflect the fact that table 1 only has demographic and medical characteristics. It does not have any social characteristics.

5. Characteristics of healthcare workers according to cadre would have added value to the findings

6. To maintain consistency with other tables, table 3 should be titled multivariate of factors associated with PACS

7. Line 299-302 is exactly the same as line 307-309. Please review.

8. Authors have captured the study limitations succinctly

Reviewer #2: This is a study of the frequency and risk factors for PACS after COVID-19 in healthcare workers. It includes important facts, such as that a booster dose of the vaccine reduces the incidence of PACS.

Major comments.

A reference to the type of vaccine that the healthcare workers in the study were vaccinated with is warranted. If unknown, this should be addressed as a limitation.

Healthcare workers with underlying asthma are considered to be at higher risk of PACS, but it should be mentioned whether the risk is higher for PACS with respiratory symptoms or for PACS as a whole, including neuropsychiatric symptoms.

Minor comment.

The resolution of the figure is low and a higher resolution figure should be provided.

6. PLOS authors have the option to publish the peer review history of their article (what does this mean?). If published, this will include your full peer review and any attached files.

Reviewer #1: No

Reviewer #2: No

---

## [Author Response · Author response to Decision Letter 0]

18 Jan 2024

Responses

Editor

1. Please ensure that your manuscript meets PLOS ONE’s style requirements, including those for file naming. The PLOS ONE style templates can be found at

We appreciate the editor’s concern regarding the manuscript’s adherence to PLOS ONE’s style requirements, including file naming. We would like to confirm that we have diligently followed these guidelines and have ensured that our manuscript complies with the specified formatting standards. We have reviewed our submission following the provided templates. If the editors have noticed any specific concerns or discrepancies, please let us know, and we will promptly address them.

We appreciate the editor’s request for additional information regarding participant consent in our study. We have included the additional details regarding the ethics statement in the submission information and our manuscript on 

Page 12, lines 212-217: “In adherence to ethical guidelines, digital informed consent was sought from all participants involved in this study. Participants who agreed to participate in this study were sent informed consent approval and details regarding this study. All respondents were assured of confidentiality. No consents were obtained from parents or guardians as our study did not involve minors.”

3. Please provide a complete Data Availability Statement in the submission form, ensuring you include all necessary access information or a reason for why you are unable to make your data freely accessible. If your research concerns only data provided within your submission, please write “All data are in the manuscript and/or supporting information files” as your Data Availability. 

Thank you, editor. We have provided a complete Data Availability Statement in the submission form and also our manuscript on 

Pages 28-29, lines 455-458: “All relevant data within the manuscript is not copyedited and is the authors’ sole responsibility. All relevant data is also within the paper and its Supporting Information files. This will ensure transparency and accessibility of data for review and replication purposes.” following the reviewer’s request. 

4. PLOS requires an ORCID iD for the corresponding author in Editorial Manager on papers submitted after December 6th, 2016. Please ensure that you have an ORCID iD and that it is validated in Editorial Manager. To do this, go to ‘Update my Information’ (in the upper left-hand corner of the main menu), and click on the Fetch/Validate link next to the ORCID field. This will take you to the ORCID site and allow you to create a new iD or authenticate a pre-existing iD in Editorial Manager. Please see the following video for instructions on linking an ORCID iD to your Editorial Manager account: https://www.youtube.com/watch?v=_xcclfuvtxQ.

Thank you, editor. ORCID iD for the corresponding author has now been included.

5. We are unable to open your Supporting Information file [Manuscript Data.sav]. Please kindly revise as necessary and re-upload.

We regret the inconvenience and appreciate your notification. We have addressed the issue with the Supporting Information file and made the necessary revisions. We have now uploaded both data formats in SPSS(SAV) and Microsoft Excel Worksheet(xlsx) format. 

We thank the editor for checking the reference list and ensuring its completeness and accuracy. We have reviewed our reference list and assured it is complete and correct. 

Reviewer: 1 

This is a well-structured manuscript which presents findings from a cross-sectional study that investigates prevalence of post-acute covid syndrome and its associated risk factors among healthcare workers in a large COVID referral hospital in Malaysia. The manuscript is simple and captures the necessary information well. It is structured accordingly and easy to read. Conclusions are sound and based on findings from the study. Findings add to the existing body of knowledge.

A couple of areas for improvement as follows:

1. There are some grammatical errors in the paper. Authors use past tense, present tense and occasional future tense throughout the paper, especially in reporting results. 

We sincerely appreciate your attention to detail and agree that maintaining a consistent tense is crucial for clarity and readability. We have carefully reviewed the entire manuscript to correct any inappropriate shifts in tense.

2. Line 91-93 sounds incomplete.

We thank the reviewers for highlighting this. We have amended the sentence at lines 91-93 previously stating “Among the most common mental effects of PACS including generalised anxiety, depression, post-traumatic stress disorder (PTSD), and sleep disturbance” to: 

Page 6, lines 99-100: “PACS is commonly associated with generalised anxiety, depression, post-traumatic stress disorder (PTSD), and sleep disturbance.”

3. Line 232 in the results section, figure 1 is referred to but it is only available as an additional resource. It would be helpful to the reader if the figure is available in the paper just like the tables.

We thank the reviewer for the valuable feedback. We have attached Figure 1 to the main text for clarity on page 14, line 239.

4. Correct line 233 to reflect the fact that table 1 only has demographic and medical characteristics. It does not have any social characteristics.

Thank you for your valuable feedback regarding the description of Table 1 in line 233 of our manuscript. We agree with your comment that the table currently only includes the demographic and medical characteristics of the participants and lacks any social characteristics. In line with your suggestion, we will revise line 233 of the manuscript to accurately reflect this. 

Page 15, lines 240-241: “The demographic and medical characteristics of infected healthcare workers with and without PACS are illustrated in Table 1.” 

5. Characteristics of healthcare workers according to cadre would have added value to the findings.

We thank you for your insightful suggestion regarding including characteristics of healthcare workers according to their cadre in our study. Unfortunately, in the design of our study, we did not collect data on the specific cadres of healthcare workers. We have now acknowledged it as one of the current study’s limitations:

Page 25, lines 419-420: “The other limitation of this study is that we did not collect data on the occupation of HCWs, which could have yielded further findings.” 

6. To maintain consistency with other tables, table 3 should be titled multivariate of factors associated with PACS

We thank you for your comments. The title for Table 3 on page 20, line 303 has now been revised to “Multivariable Analysis of Factors Associated with PACS.” 

7. Line 299-302 is exactly the same as line 307-309. Please review.

We thank you for pointing out the duplication in lines 299-302 and 307-309 of our manuscript. We have now removed lines 299-302 from the manuscript to rectify this. This ensures the content is presented without repetition, improving our paper’s clarity and conciseness.

8. Authors have captured the study limitations succinctly. 

We thank the reviewer for the positive feedback. We have added two additional limitations as stated below:

Page 25, lines 418-419: “Moreover, we did not collect data on the type of vaccine HCWs took, which could produce different levels of protection against PACS.”

Page 25, lines 419-420: “The other limitation of this study is that we did not collect data on the occupation of HCWs, which could have yielded further findings.”

 

Reviewer: 2 

This is a study of the frequency and risk factors for PACS after COVID-19 in healthcare workers. It includes important facts, such as that a booster dose of the vaccine reduces the incidence of PACS.

Major comments

1. A reference to the type of vaccine that the healthcare workers in the study were vaccinated with is warranted. If unknown, this should be addressed as a limitation.

Healthcare workers with underlying asthma are considered to be at higher risk of PACS, but it should be mentioned whether the risk is higher for PACS with respiratory symptoms or for PACS as a whole, including neuropsychiatric symptoms.

We thank the reviewer for highlighting this. We acknowledge the importance of providing information about the type of vaccine administered to healthcare workers in our study. Unfortunately, this information was not available to us, and we understand that it is a limitation of our research. We have added this limitation to our manuscript on 

Page 25, lines 418-419: “Moreover, we did not collect data on the type of vaccine HCWs took, which could produce different levels of protection against PACS.”

Regarding the second comment about healthcare workers with underlying asthma and their risk of Post-Acute COVID-19 Syndrome (PACS):

We appreciate the reviewer’s comment regarding the risk of PACS in healthcare workers with underlying asthma. Our study primarily focused on the risk of PACS as a whole, including both respiratory and neuropsychiatric symptoms. We did not specifically differentiate between the two within this study. To address this, we have amended.

Page 23, lines 369-370: “Among the comorbidities analysed, asthma is the only significant risk factor for developing PACS in general, irrespective of symptoms, with an aOR of 2.01”

Minor comment.

2. The resolution of the figure is low and a higher resolution figure should be provided.

We appreciate the reviewer’s feedback regarding the figure resolution. We have taken the previous reviewer’s suggestion and included a clearer figure in the manuscript.

---

## [Editor Report · Decision Letter 1]

24 Jan 2024

Prevalence and Predictors of Post-Acute COVID Syndrome Among Infected Healthcare Workers at University Malaya Medical Centre

PONE-D-23-35985R1

Dear Dr. Lim,

We’re pleased to inform you that your manuscript has been judged scientifically suitable for publication and will be formally accepted for publication once it meets all outstanding technical requirements.

Kind regards,

Belal Nedal Sabbah

Academic Editor

PLOS ONE
---

## [Editor Report · Acceptance letter]

2 Apr 2024

PONE-D-23-35985R1 

PLOS ONE

Dear Dr. Lim, 

I'm pleased to inform you that your manuscript has been deemed suitable for publication in PLOS ONE. Congratulations! Your manuscript is now being handed over to our production team.

Kind regards, 

on behalf of

Dr. Belal Nedal Sabbah 

Academic Editor

PLOS ONE